# The Influence of Friends on Teen Vaping: A Mixed-Methods Approach

**DOI:** 10.3390/ijerph18136784

**Published:** 2021-06-24

**Authors:** Allison L. Groom, Thanh-Huyen T. Vu, Robyn L. Landry, Anshula Kesh, Joy L. Hart, Kandi L. Walker, Lindsey A. Wood, Rose Marie Robertson, Thomas J. Payne

**Affiliations:** 1American Heart Association Tobacco Regulation and Addiction Center, Dallas, TX 75231, USA; robyn.landry@heart.org (R.L.L.); a.kesh@heart.org (A.K.); rosemarie.robertson@heart.org (R.M.R.); 2Department of Preventive Medicine, Feinberg School of Medicine, Northwestern University, Chicago, IL 60611, USA; huyenvu@northwestern.edu; 3Department of Communication, University of Louisville, Louisville, KY 40292, USA; joy.hart@louisville.edu (J.L.H.); kandi.walker@louisville.edu (K.L.W.); lindsey.wood.1@louisville.edu (L.A.W.); 4Department of Medicine, Vanderbilt University Medical Center, Nashville, TN 37232, USA; 5Department of Otolaryngology, Head and Neck Surgery, University of Mississippi Medical Center, Jackson, MS 39216, USA; drtjp123@gmail.com; 6Department of Psychiatry, Memorial Sloan Kettering Cancer Center, New York, NY 10065, USA

**Keywords:** vaping, electronic nicotine delivery systems, teen, adolescent, peers

## Abstract

Vaping is popular among adolescents. Previous research has explored sources of information and influence on youth vaping, including marketing, ads, family, peers, social media, and the internet. This research endeavors to expand understanding of peer influence. Our hypothesis is that friends’ influence on teen vapers’ first electronic nicotine delivery systems (ENDS) use varies by demographic variables and awareness of ENDS advertising. In August–October 2017, youth (*n =* 3174) aged 13–18 completed an online survey to quantify ENDS behaviors and attitudes and were invited to participate in follow-up online research in November-December 2017 to probe qualitative context around perceptions and motivations (*n =* 76). This analysis focused on the ENDS users, defined as having ever tried any ENDS product, from the survey (*n =* 1549) and the follow-up research (*n =* 39). Among survey respondents, friends were the most common source of vapers’ first ENDS product (60%). Most survey respondents tried their first ENDS product while “hanging out with friends” (54%). Among follow-up research participants, the theme of socializing was also prominent. ENDS advertising and marketing through social media had a strong association with friend networks; in fact, the odds of friends as source of the first vaping experience were 2 times higher for those who had seen ENDS ads on social media compared with other types of media. The influence of friends is particularly evident among non-Hispanic Whites, Hispanics/Latinos, those living in urban areas, those living in high-income households, those with higher self-esteem, and those who experiment with vaping. These findings support the premise that peer influence is a primary social influencer and reinforcer for vaping. Being included in a popular activity appears to be a strong driving force.

## 1. Introduction

Awareness and use of electronic cigarettes (e-cigarettes) and other electronic nicotine delivery systems (ENDS) has become increasingly pervasive among U.S. teens in recent years. High rates of e-cigarette awareness have been observed among middle school (84.3%) and high school (92.0%) students [1,2]. Further, in the 2019 National Youth Tobacco Survey, 27.5% of high school students and 10.5% of middle school students reported current e-cigarette use. More than five million students had used e-cigarettes in the past 30 days; nearly one million used them daily [3,4]. More than half (51.2%) of middle school e-cigarette users identified e-cigarettes as the first tobacco product tried [1].

Teen vaping can be influenced by several sources of information, including marketing, family, peers, and the internet. Exposure to e-cigarette marketing and advertising is associated with openness to using e-cigarettes and curiosity about trying e-cigarettes [5] and predicts subsequent e-cigarette experimentation among teens who have never used tobacco [6]. Furthermore, exposure to e-cigarette marketing is associated with use of other tobacco products, including cigarettes, hookah and cigars, as well as polytobacco product use [7]. Exposure to tobacco product promotions, including advertising and social media, is significantly associated with ever and current smoking and vaping, and susceptibility to vaping among never-users [8].

Vaping is popular among teens. Both teens who had used and those who had never used e-cigarettes acknowledged their popularity and acceptance among their peers [9]. The literature on teen vaping identifies peer influence (having peers who use tobacco) as one of the most common drivers of teen e-cigarette use, with demographics (male gender identity, older age, higher amount of pocket money) and other tobacco use behavior (such as regular and heavier use) also associated with use [10]. Peer influence (friends use) is cited as a reason for using ENDS [11] and, in particular, for liking JUUL pods [12]. Social acceptance (cool social image, fitting in) plays a role in appeal [13]. The most common source for getting e-cigarettes is a friend, followed by a family member. A quarter of youth live with someone who uses e-cigarettes, which plays a role in vaping: a third of youth who live with an e-cigarette user reported receiving or buying e-cigarettes from a family member, a higher proportion compared to those not living with an e-cigarette user [14].

This research endeavors to expand understanding of influences on teen vaping. Our hypothesis is that friends’ influence on teen vapers’ first ENDS use varies by demographic variables and awareness of ENDS advertising.

## 2. Materials and Methods

A mixed-methods approach was used, including online quantitative and qualitative analyses. This approach allowed us to quantify vaping behavior, knowledge, and attitudes, as well as explore motivations and barriers through in-depth responses.

### 2.1. Quantitative Online Survey

#### 2.1.1. Recruitment

From August to October 2017, researchers conducted a quantitative online survey with a U.S. sample of teens aged 13–18. The online approach provided access to a diverse nationwide sample, recruited by an established marketing research vendor that manages an online panel of 65,000 U.S. teens and young adults. Members were recruited via buzz campaigns, newspaper ads, and social networks. Panelists earn points for each completed survey that can be redeemed for prizes. Panel management procedures complied with marketing research industry standards set by professional marketing research associations. Procedures for obtaining proper online consent were implemented. No identifying information was collected, and guidelines established by the Children’s Online Privacy Protection Act (COPPA) were followed. Teen participants were given assent forms and could elect not to participate. Parental consent was obtained for panelists under the age of 18; parents and children were informed that parents would have no access to study data. The study team had no direct contact with recruited individuals. The Chesapeake/Advarra Institutional Review Board reviewed and approved this study [15,16,17,18].

#### 2.1.2. Sample

The study sample consisted of 3174 participants. The inclusion criteria were based on ENDS use status (users and non-users), age, sex, race/ethnicity, and nationwide geographic representation. Two groups of U.S. youth aged 13 to 18 years were recruited: (a) ENDS users, defined as teens who have ever tried e-cigarettes or other ENDS and (b) a control group, defined as teens who have never tried ENDS. This analysis focused on ENDS users (*n =* 1549). Although respondents were asked about their current ENDS use, the focus of this research was on initiation; therefore, all respondents who had ever tried ENDS were included in the analysis. Quotas were set for key demographics, ensuring sufficient numbers of participants to examine or control for the following factors: age, sex, and race/ethnicity. Non-Hispanic Black and Hispanic/Latino respondents were oversampled to ensure sufficient sample sizes for comparison by race and ethnicity. Age, sex, and race/ethnicity data were employed to accurately weight the results. The data were weighted to be representative of the overall U.S. population in terms of age, sex, race, ethnicity, and region, based on U.S. Census data [15,16,17,18].

#### 2.1.3. Measures

Demographic variables included age group based on birth year and month, sex, sexual orientation (straight or lesbian/gay/bisexual/transgender/queer), race/ethnicity (non-Hispanic White, non-Hispanic Black, Hispanic/Latino, and non-Hispanic Other--including more than one race, Asian or Pacific Islander, Native American/Alaska Native), place of residence (urban, suburban, or rural), and household income. Household income status was categorized as low vs. high, with low-income status defined as the respondent participating in a free/reduced cost lunch program at school or family receiving government/public assistance (Medicaid, Section 8 housing, Obama phone, food stamps, the link card/SNAP, or other government financial help). To determine ever-ENDS use, we asked “Which of the following types of tobacco have you ever tried (even one time or two times)?” and listed a choice of 10 tobacco product types with corresponding images: (1) electronic nicotine products, (2) traditional cigarettes, (3) traditional cigars, (4) cigarillos, (5) smokeless tobacco, (6) hookahs to smoke tobacco, (7) little or filtered cigars, (8) dissolvable tobacco products, (9) bidis and/or kreteks, and (10) others [15,16,17,18]. Vaping status was categorized as current (within the last 30 days), experimental (occasionally, but less than monthly), and former (in the past, but not now). We asked two questions regarding exposure to ENDS advertising: “In the past 3 months, have you heard, seen, or read advertising for electronic nicotine products?” and “Where have you heard, read, or seen advertising or marketing for electronic nicotine products? Choose all that apply.” Respondents were asked to choose from a list of 25 items and eight of these 25 items (i.e., Facebook, Instagram, Snapchat, Twitter, Tumblr, Pinterest, Periscope, and Bumble) were categorized as social media vs. non-social media. To examine influencers, we asked “Where did you get your first electronic nicotine product? Choose one.” Respondents were shown a list of options, the order of which was randomized, and were asked to choose one of the following: (1) a friend, (2) a family member or relative, (3) a neighbor, (4) someone else, but not a friend or relative, (5) I bought it at a store, (6) other, and (7) I don’t remember. Respondents were then asked “Where were you when you first used electronic nicotine products? Choose one.” They were asked to choose one of the following randomized answers: (1) hanging out with friends, (2) at parties, (3) by myself, (4) with my family, (5) school, (6) other, and (7) I don’t remember. Finally, we assessed self-esteem by asking “Self-esteem is defined as how much you like yourself. Please respond to the following statement: “I have high self-esteem.” Respondents were asked to rate their self-esteem on a 7-point Likert scale where 1 = not very true of me and 7 = very true of me. The responses were grouped into two categories: low (1–4) and high (5–7) [19,20].

#### 2.1.4. Statistical Analysis

Descriptive analyses were used to show the distribution of where respondents got their first ENDS product and where they first used ENDS. Differences in demographic characteristics related to influences were compared using Chi-square tests. In multivariable analyses, logistic regression models were used to estimate the odds of reporting a friend as the source of first ENDS product as well as reporting hanging out with friends as the location of first ENDS use by age group, sex, race/ethnicity categories, place of residence, household income status, sexual orientation, awareness of ENDS advertising or marketing, vaping status, and self-esteem. Sampling weight was generated and applied in the analysis. Analyses were conducted with SAS statistical software (version 9.4 with SAS/STAT 14.1, SAS Institute Inc., Cary, NC, USA).

### 2.2. Qualitative Online Community Research

#### 2.2.1. Recruitment

Survey respondents were invited to participate in a follow-up online community. During two weeks in November and December 2017, participants were asked to visit the online community each day to answer questions and conduct interactive activities.

#### 2.2.2. Sample

A total of 76 survey respondents participated in the online community, including 39 ENDS users and 37 never ENDS users. This analysis focused on the 39 ENDS users.

#### 2.2.3. Measures

The participants were asked to “Finish the following sentences to help us learn more about vaping: I vape because_____; The best things about vaping are _____; The worst things about vaping are_____.” They were allowed to enter multiple responses.

#### 2.2.4. Analysis

The data were analyzed using inductive qualitative content analysis to identify themes that emerged. Using an open coding technique, codes were assigned to participant responses using their words and uploaded images to establish the coding scheme.

## 3. Results

### 3.1. Quantitative Online Survey

#### 3.1.1. Sample Characteristics

The weighted sample of 1549 ENDS users included teens across three age groups: 13–14 (14.1%), 15–16 (34.7%), and 17–18 (51.3%). Of the sample, 56.9% were male and 43.1% were female; 64.4% were non-Hispanic White, 10.1% were non-Hispanic Black, and 3.7% were non-Hispanic Other; 21.7% were Hispanic/Latino; 22.7% identified as LGBTQ; 52.8% were from low-income households; 36% lived in urban areas, 40.2% suburban, and 23.8% rural; and 24.5% rated their self-esteem as low. Additionally, in terms of vaping status, 35% were current users, 21.3% experimenters, and 43.7% former users. We observed significant differences by sex with regard to age (females skewed older than males), race/ethnicity (females were more apt than males to be non-Hispanic White), sexual orientation (females were more apt than males to identify as LGBTQ), vaping status (females were more likely to be experimenters or former users), and self-esteem (females were more likely to rate their self-esteem as low). (Table 1)

#### 3.1.2. Awareness of ENDS Advertising

Slightly more than one-half of the ENDS users (*n =* 830, 53.6%) in the online survey said they had heard, seen, or read advertising for electronic nicotine products in the past three months, and the majority (*n =* 508) received information from social media.

The most common sources of advertising were point-of-purchase outlets: vape stores (48.6%) and convenience stores or gas stations (41.0%). Other common sources were TV (38.1%), Facebook (33.0%), Instagram (31.9%), YouTube video (31.4%), and website (25.2%) (Table 2).

#### 3.1.3. Sources of First ENDS Product

Among survey respondents, friends were the most common source of the first ENDS product (59.7%). Less frequently cited sources were a family member or relative (16.0%), store (8.0%), someone else/other (9.0%), and don’t remember (7.3%). Significant differences were observed by age, sex, race, awareness of ENDS advertising/marketing, and self-esteem. Older teens were significantly more likely than younger teens to identify a friend as the source of their first ENDS product (62.3% of 17–18 vs. 57.5% of 13–14 and 56.8% of 15–16, *p*-Value < 0.016). Older teens were significantly *less* likely than younger teens to identify a family member as the source (12.9% of 17–18 vs. 18.8% of 13–14 and 19.5% of 15–16, *p*-Value < 0.016). Although friends were the most common source for both females and males, females were significantly more likely than males to indicate a family member was the source (20.9% vs. 12.3%, *p*-Value < 0.001) (Table 3).

The odds of ENDS users identifying a friend as the source of their first ENDS product were significantly higher for non-Hispanic Whites compared with non-Hispanic Blacks (OR: 1.78, 95% CI: 1.19, 2.67; *p* = 0.005) and Hispanics/Latinos compared with non-Hispanic Blacks (OR: 1.95, 95% CI: 1.19, 3.20; *p* = 0.008); users who had seen, heard, or read ENDS advertising on social media compared with other types of media channels or locations (OR: 2.04, 95% CI: 1.41, 2.96; *p <* 0.001); users with high self-esteem compared with users with low self-esteem (OR: 1.35, 95% CI: 1.02, 1.78; *p =* 0.038); urban compared with rural (OR: 1.50, 95% CI: 1.04, 2.15; *p =* 0.028); high income compared with low income (OR: 1.97, 95% CI: 1.49, 2.61; *p <* 0.001); and vaping experimenters (OR: 1.80, 95% CI: 1.25, 2.61; *p <* 0.002) (Table 4).

#### 3.1.4. Location of First ENDS Product Use

Most online respondents tried their first ENDS product while “hanging out with friends” (54.0%). Less frequent locations reported were by myself (13.5%), with my family (10.1%), at school (7.8%), at parties (7.1%), I don’t remember (5.5%), and other (2.1%). Significant differences were observed by sex and race/ethnicity. Although friends were mentioned most often by both males and females (55.4% of females; 52.9% of males), females were significantly *more* likely than males to have tried their first ENDS product with family (13.9% of females vs. 7.3% of males, *p*-value < 0.001). Non-Hispanic White and Hispanic/Latino respondents were significantly more likely than non-Hispanic Black respondents to have tried their first ENDS product with friends (56.7% of non-Hispanic White and 54.6% of Hispanic/Latino vs. 37.7% of non-Hispanic Black, *p*-value = 0.042). (Table 5) The odds of ENDS users trying their first ENDS product while “hanging out with friends” were significantly *higher* for non-Hispanic Whites compared with non-Hispanic Blacks (OR: 2.04, 95% CI: 1.35, 3.06; *p <* 0.001) and Hispanics/Latinos compared with non-Hispanic Blacks (OR: 1.91, 95% CI: 1.17, 3.10; *p =* 0.010). The odds were significantly lower for the 13–14 age group compared with the 17–18 age group (OR: 0.67, 95% DI: 0.44, 1.02; *p =* 0.062). The odds were significantly higher for high income compared with low income respondents (OR: 1.60, 95% CI: 1.22, 2.11; *p <* 0.001). And, the odds were signficantly higher for those who had seen ENDS ads on social media compared with other types of media (OR: 1.51, 95% CI: 1.04, 2.18; *p =* 0.031) and for vaping experimenters (OR: 1.56, 95% CI: 1.08, 2.27; *p =* 0.019) (Table 6).

### 3.2. Qualitative Online Community Research

#### 3.2.1. Sample Characteristics

The demographic subgroups in the sample of 39 ENDS users were collapsed due to the smaller sample size. The subgroups included two age groups: 13–15 (17.9%) and 16–18 (82.1%); 41.0% were male, 56.4% were female; and one participant identified as non-binary who had not done so in the online survey; 51.3% were non-Hispanic White and 48.7% were Other (non-Hispanic Black, non-Hispanic Other, Hispanic/Latino) (Table 7).

#### 3.2.2. Sources of First ENDS Product

Many of the online community participants reported that, shortly after they first tried vaping, they made vaping purchases of their own for the first time. A majority bought products through a friend, a friend’s older sibling, or a friend of a friend, who guided them through the process. These more experienced vapers advised them about flavors and equipment, and sometimes made the purchase for them.

#### 3.2.3. Associations with Vaping

In the Qualitative Online Community, the theme of socializing was prominent: one of the reasons they vaped the first time was because “my friends and I do it together” and one of the best things about vaping was “socializing with friends.” (Figure 1) Most of the time ENDS users vaped with friends or other people their age. Together they shared flavors and exchanged liquids to experiment. Most vapers indicated they would vape less if their friends didn’t vape. Many participants conveyed the importance of peer influence with quotes such as the following:


*“I know a lot of people who vape. It’s pretty popular with most of my friends. If my friends didn’t vape, I doubt I would have ever started.”*
(Male, 16–18)


*“My friends and I vape together almost every time we are together. If they did it less I probably would too.”*
(Female, 13–15)

Although teens tend to be open with friends about their vaping, some acknowledged the opinions of their non-vaping friends and reported a stigma against vaping that prevents them from vaping openly. (Figure 2) One of the worst things about vaping, according to some ENDS users, is the negative stigma:


*“People are quick to judge you if you do it.”*
(Female, 13–15) (Figure 1)

## 4. Discussion

Our findings provide evidence of the important role friends play in the lives of most teens and their decision to start vaping. Most teens get their first ENDS product from a friend and recount that their first vaping experience was with friends. Older friends and acquaintances play an advisory role to the newly initiated vaper. The influence of friends is particularly evident among non-Hispanic Whites, Hispanics/Latinos, those living in urban areas, those living in high-income households, those with high self-esteem, and those who experiment with vaping. Although friends are also the most common influence among non-Hispanic Blacks, family played a more substantial role relative to other race groups. The greater influence of family was also observed for females compared with males.

ENDS advertising and marketing through social media has a strong association with friend networks, reflected by the fact that ENDS users who were initially influenced by a friend were more likely to have been exposed to ENDS messaging on social media. Acknowleding the prominent role that social media play in many teens’ social networks, anti-vaping ads placed in social media channels used by teens may have the potential to offset peer influence.

The insights from the qualitative research highlight the role of friends in the vaping experience, with some saying they might not have started vaping if their friends didn’t vape, or they might vape less if their friends vaped less. It is possible that greater time spent on social media and physical distancing practices during the COVID-19 pandemic may influence vaping behavior; these are important topics for future research.

Vaping may have a negative social stigma among friends who do not vape. Some ENDS users mentioned hiding their vaping habit from friends, although most felt comfortable with vaping openly and did not convey a sense of shame. Exploring the influence that non-vapers might have on their friends could be useful in understanding how to dissuade teen vaping initiation and uptake or encourage and support cessation.

The research had some limitations. Survey respondents who opted into the follow-up community may not represent the full survey sample or general population of teens who vape. Also, this research was conducted prior to the COVID-19 pandemic, so behavior reported does not reflect the physical distancing practiced by some teens during the pandemic or potential changes in access to e-cigarette products. Additionally, new vaping products have appeared in the marketplace since this research was conducted. Finally, this analysis focused on social aspects of vaping rather than the adverse health effects of vaping [21,22,23] and perceptions of health consequences [16], an important topic for future analysis.

## 5. Conclusions

These findings support the premise that peers are a primary social influencer and reinforcer for vaping. Inclusion in a popular activity appears to be a strong driving force among teens in general, but particularly among older teens (17–18), males, and non-Hispanic White and Hispanic/Latino teens. These findings support previous research indicating that certain demographics are more susceptible to peer influence than others and add new insight about the impact of social media and friends as the source of initial product use. Educational and social media strategies should consider the importance of peer influence, as well as that of the family. The retail environment is also a notable source, indicating the need for increased enforcement of purchase age restrictions.

Future research on peer influence could expand knowledge by examining: (1) non-vapers’ influence on preventing friends from vaping, (2) the potential of substituting alternative popular activities, and (3) the impact of physical distancing, stay-at-home policies and remote learning during the COVID-19 pandemic.

## Figures and Tables

**Figure 1 ijerph-18-06784-f001:**
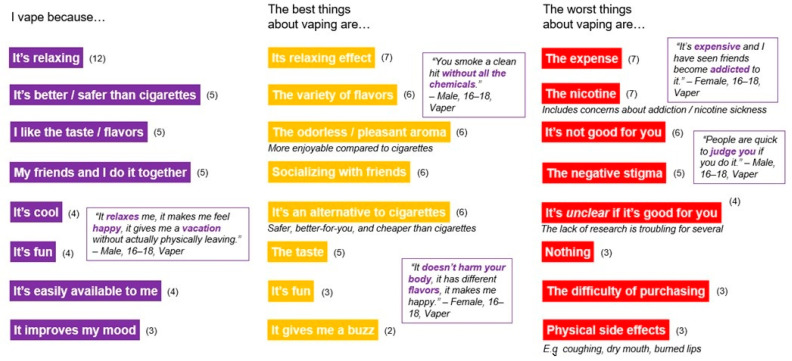
Associations with Vaping. Note: Numbers in parentheses represent the number of participant comments related to the theme.

**Figure 2 ijerph-18-06784-f002:**
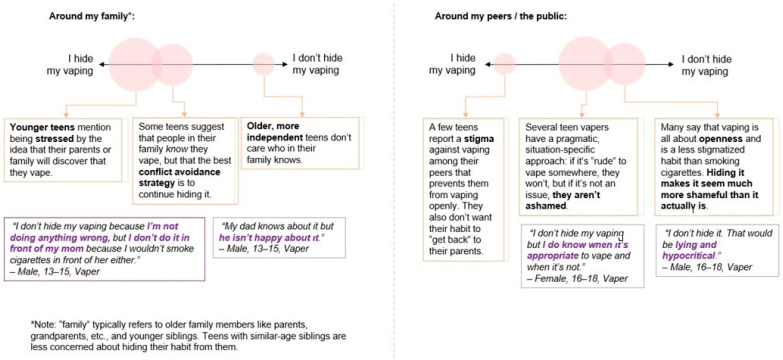
Openness About Vaping. Note: The size of the bubbles corresponds with the proportion of participant comments related to the theme.

**Table 1 ijerph-18-06784-t001:** Participant Characteristics, Overall and by Sex—Quantitative Survey.

Characteristics	*n*	%	Male (%)	Female (%)	*p*-Value *
	1549				
**Sex**					NA
Male	882	56.9	100.0		
Female	668	43.1		100.0	
**Race/Ethnicity**					**0.042**
Non-Hispanic White	998	64.4	62.7	66.7	
Non-Hispanic Black	157	10.1	9.0	11.6	
Non-Hispanic Other	57	3.7	3.7	3.7	
Hispanic/Latino	336	21.7	24.5	18.0	
**Age Group, year**					**0.023**
13–14	218	14.1	14.7	13.3	
15–16	537	34.7	38.2	30.0	
17–18	794	51.3	47.2	56.6	
**Sexual Orientation**					**<0.001**
Lesbian/bisexual/gay/transgender/queer (LBGTQ)	351	22.7	13.4	35.0	
Straight	1198	77.3	86.6	65.0	
**Residence**					0.145
Urban	558	36.0	38.4	32.8	
Suburban	623	40.2	39.7	40.9	
Rural	368	23.8	21.9	26.2	
**Income Status**					0.253
Low income **	818	52.8	48.8	45.0	
**Vaping Status**					**<0.001**
Current	542	35.0	44.1	22.9	
Experimenter	331	21.3	19.8	23.4	
Former	676	43.7	36.1	53.7	
**Self-esteem**					**<0.001**
Low (rated 1–4)	640	41.3	31.5	54.3	
High (rated 5–7)	909	58.7	68.5	45.7	

Data are weighted. * *p*-Value for comparisons between male and female participants based on Rao Scott χ^2^ tests. ** Low-income status defined as participating in a free/reduced cost lunch program at school or family receiving government public assistance (Medicaid, Section 8 housing, Obama phone, food stamps, the link card/SNAP, or other government financial help). The bold numbers represent statistical significance.

**Table 2 ijerph-18-06784-t002:** Awareness of ENDS advertising or marketing—Quantitative Survey.

Online Survey	*N*	%
Any Aware of Advertising/Marketing	830	53.6
Any Social Media (net) *	508	32.8
Any Other Type of Media/Locations (net) **	321	20.8
Specific Types of Advertising/Marketing (select all that apply)		
1. Vape stores **	403	48.6
2. Convenience stores or gas stations **	340	41.0
3. TV **	316	38.1
4. Facebook *	274	33.0
5. Instagram *	264	31.9
6. YouTube video **	261	31.4
7. Website **	209	25.2
8. Snapchat *	191	23.1
9. Billboards **	146	17.6
10. Twitter *	144	17.4
11. Newspaper or magazine **	135	16.2
12. Radio **	124	14.9
13. Tumblr *	80	9.6
14. Email **	73	8.8
15. Pinterest *	58	7.0
16. Netflix **	52	6.3
17. Kik **	32	3.9
18. Skype **	27	3.2
19. Hulu **	26	3.1
20. Periscope *	23	2.7
21. Other **	21	2.6
22. Dailymotion **	21	2.5
23. What’s App **	17	2.0
24. Bumble *	14	1.7
25. Vimeo **	12	1.4

Data are weighted. * Social media sources. ** Other types of media sources/locations.

**Table 3 ijerph-18-06784-t003:** Source of First ENDS Product, Overall and by Sex, Age, and Race/Ethnicity—Quantitative Survey.

Sources	Overall		Sex		Age			Race/Ethnicity			
Chi-Square *	-	-	30.1544		24.8362			24.2902			
*p*-Value *	-	-	**<0.001**		**0.016**			0.146			
**Location**	*N*	Total	Female	Male	13–14	15–16	17–18	NH White	NH Black	Hispanic	NH Other
*N*	1549	%	668	882	218	537	794	998	157	336	57
		%	%	%	%	%	%	%	%	%	%
A friend	925	59.7	58.4	60.7	57.5	56.8	62.3	60.3	47.1	63.6	62.3
A family member or relative	248	16.0	20.9	12.3	18.8	19.5	12.9	16.5	20.4	12.7	14.2
A neighbor	34	2.2	0.9	3.2	3.5	2.6	1.6	2.2	2.3	1.9	4.9
Someone else, but not a friend or relative	68	4.4	3.5	5.1	2.2	5.4	4.3	3.3	5.5	7.3	3.7
I bought it at a store	124	8.0	5.1	10.2	3.7	6.1	10.5	7.8	8.5	8.8	5.5
Other	37	2.4	2.3	2.4	2.7	1.7	2.7	2.1	4.0	2.3	1.9
I don’t remember	113	7.3	8.9	6.1	11.5	8.0	5.7	7.9	12.1	3.5	7.6

* *p*-Value for comparisons across source of first ENDS product based on Rao Scott χ^2^ tests. Data are weighted.

**Table 4 ijerph-18-06784-t004:** Multivariable Adjusted Odds Ratios (95% CI) of Friends as Source of First ENDS Product – Quantitative Survey.

Characteristics	OR (95% CI)	*p*-Value
**Age Group, year**		
Age 13–14 vs. 17–18	0.86 (0.56, 1.32)	0.486
Age 15–16 vs. 17–18	0.80 (0.59, 1.07)	0.135
**Girls vs. Boys**	0.97 (0.72, 1.31)	0.846
**Race/Ethnicity**		
Hispanic vs. NH-Black	**1.** **95 (1.19, 3.20** **)**	**0.008**
NH-White vs. NH-Black	**1.** **78 (1.19, 2.67)**	**0.005**
NH-Other vs. NH-Black	**1.** **91 (1.06, 3.45)**	**0.032**
**Residence**		
Urban vs. Rural	**1.50 (1.04, 2.15)**	**0.028**
Suburban vs. Rural	1.27 (0.89, 1.81)	0.183
**High Income vs. Low Income ***	**1.97 (1.49, 2.61)**	**<0.001**
**Sexual Orientation Straight vs. LGBTQ**	0.99 (0.71, 1.36)	0.928
**ENDS Advertising/Marketing**		
Any Social Media vs. Non-Social Media **	**2.** **04 (1.41, 2.96)**	**<0.001**
Never Heard vs. Non-Social Media	**1.** **71 (1.21, 2.43)**	**0.003**
**High Self-Esteem vs. Low Self-Esteem *****	**1.** **35 (1.02, 1.78)**	**0.038**
**Vaping Status**		
Experimenter vs. Current User	**1.80 (1.25, 2.61)**	**0.002**
Former vs. Current User	1.19 (0.88, 1.62)	0.264

* Low-income status defined as participating in a free/reduced cost lunch program at school or family receiving government public assistance (Medicaid, Section 8 housing, Obama phone, food stamps, the link card/SNAP, or other government financial help); ** Social media sources—Facebook, Instagram, Snapchat, Twitter, Tumblr, Pinterest, Periscope, and Bumble; *** High self-esteem, rated 5–7; low self-esteem, rated 1–4.

**Table 5 ijerph-18-06784-t005:** Location of First ENDS Product Use, Overall and by Sex, Age, and Race/Ethnicity—Quantitative Survey.

Locations	Overall		Sex		Age Group			Race/Ethnicity			
Chi-Square *			23.8982		15.2037			29.5828			
*p*-Value *			**<0.001**		0.231			**0.042**			
Location	*N*	Total	Female	Male	13–14	15–16	17–18	NH White	NH Black	Hispanic	NH Others
*N*	1549		668	882	218	537	794	998	157	336	57
		%	%	%	%	%	%	%	%	%	%
Hanging out with friends	837	54.0	55.4	52.9	45.9	51.9	57.6	56.7	37.7	54.6	48.3
At parties	110	7.1	4.5	9.1	8.5	7.8	6.2	6.1	6.9	9.1	14.2
By myself	209	13.5	11.0	15.4	11.0	14.0	13.9	13.4	19.4	11.0	14.0
With my family	156	10.1	13.9	7.3	15.4	10.8	8.2	10.4	13.5	8.2	6.9
School	120	7.8	6.6	8.6	6.3	8.3	7.8	5.9	11.1	11.4	10.1
Other	32	2.1	2.8	1.5	4.8	1.9	1.4	2.0	2.6	2.0	1.6
I don’t remember	85	5.5	5.8	5.2	8.1	5.4	4.8	5.6	8.8	3.6	5.0

* *p*-Value for comparisons across location of first ENDS use based on Rao Scott χ^2^ tests; Data are weighted.

**Table 6 ijerph-18-06784-t006:** Multivariable Adjusted Odds Ratios (95% CI) of Hanging Out with Friends as Location of First ENDS Product Use—Quantitative Survey.

Characteristics	OR (95% CI)	*p*-Value
**Age Group, year**		
Age 13–14 vs. 17–18	0.67 (0.44, 1.02)	0.062
Age 15–16 vs. 17–18	0.83 (0.62, 1.12)	0.225
**Girls vs. Boys**	1.11 (0.83, 1.50)	0.478
**Race/Ethnicity**		
Hispanic vs. NH-Black	**1.** **91 (1.17, 3.10)**	**0.010**
NH-White vs. NH-Black	**2.** **04 (1.35, 3.06)**	**0.001**
NH-Other vs. NH-Black	1.40 (0.77, 2.56)	0.274
**Residence**		
Urban vs. Rural	1.08 (0.75, 1.54)	0.694
Suburban vs. Rural	1.25 (0.87, 1.79)	0.228
**High Income vs. Low Income ***	**1.60 (1.22, 2.11)**	**0.001**
**Sexual Orientation Straight vs. LGBTQ**	0.77 (0.55, 1.07)	0.116
**ENDS Advertising/Marketing**		
Any Social Media ** vs. Non-Social Media	**1.51 (1.04, 2.18)**	**0.031**
Never Heard vs. Non-Social Media	1.22 (0.86, 1.72)	0.268
**High Self-Esteem vs. Low Self-Esteem *****	1.20 (0.91, 1.59)	0.193
**Vaping Status**		
Experimenter vs. Current User	**1.56 (1.08, 2.27)**	**0.019**
Former vs. Current User	0.91 (0.68, 1.23)	0.553

* Low-income status defined as participating in a free/reduced cost lunch program at school or family receiving government public assistance (Medicaid, Section 8 housing, Obama phone, food stamps, the link card/SNAP, or other government financial help); ** Social media sources—Facebook, Instagram, Snapchat, Twitter, Tumblr, Pinterest, Periscope, and Bumble; *** High self-esteem, rated 5–7; low self-esteem, rated 1–4.

**Table 7 ijerph-18-06784-t007:** Participant Characteristics—Qualitative Research.

Online Community	*N*	%
**ENDS Users:**	39	
**Gender**		
Male	16	41.0
Female	22	56.4
Non-binary *	1	2.6
**Race/Ethnicity**		
Non-Hispanic White	20	51.3
Other (non-Hispanic Black, non-Hispanic Other, Hispanic/Latino)	19	48.7
**Age**		
13–15	7	17.9
16–18	32	82.1

* One online community participant identified as non-binary who had not done so in the online survey.

## Data Availability

Data available upon request.

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
