# Peer review of "The Influence of Friends on Teen Vaping: A Mixed-Methods Approach"

_ijerph, 2021, doi:10.3390/ijerph18136784_

Round 1

Reviewer 1 Report

The authors should improve the presentation of table 5. 

Author Response

Thank you for reviewing the revised manuscript and for your suggestion regarding Table 5. We have reformatted the table to fit vertically on the page.

Reviewer 2 Report

The authors have addressed all of my concerns and the manuscript has improved. Prior to publication, my only recommendation is to include the authors' definition of "current" "experimental" and "former" ENDS users for context (page 3, line 124).

Author Response

Thank you for reviewing the revised manuscript and for suggesting clarification of the user definitions in the Measures section. We have revised the sentence as follows: Vaping status was categorized as current (within the last 30 days), experimental (occasionally, but less than monthly), and former (in the past, but not now). 

Reviewer 3 Report

I have no further querries

Author Response

Thank you for your review.

This manuscript is a resubmission of an earlier submission. The following is a list of the peer review reports and author responses from that submission.

Round 1

Reviewer 1 Report

The paper “The Influence of Friends on Teen Vaping: A Mixed-methods Approach” had important results but I consider that requires some clarification.

Lines 138-140. There were insufficient numbers of…

Can authors explain the sample size obtained?

Lines 160-161. It is the same question as to the previous point.

Table 1a. Authors employed race and ethnicity, please, can explain these concepts in your research?

This type of device is more expensive than conventional cigarettes.

Did you consider the socio-economic level of the study population?

Author Response

Thank you for the comments on our work. We appreciate your time and insights, and we believe that the draft has improved through the revision process. Below is a point-by-point response to the review comments, which addresses the ways that we revised the manuscript. Again, thank you for assisting us with improving the article.

Comment: The paper “The Influence of Friends on Teen Vaping: A Mixed-methods Approach” had important results but I consider that requires some clarification.

Lines 138-140. There were insufficient numbers of…Can authors explain the sample size obtained?

Lines 160-161. It is the same question as to the previous point.

Response: Thank you for pointing out the potential confusion about our reference to insufficient numbers of Asian or Pacific Islanders, Native American/Alaska Natives or mixed-race individuals (combined as non-Hispanic Other respondents). In the quantitative survey, we set quotas for non-Hispanic Black and Hispanic respondents, the largest populations related to race and ethnicity to ensure sufficient sample sizes for analysis. We did not set quotas for non-Hispanic Other respondents, and were not able to obtain a sample large enough for certain analyses. For the qualitative research, we anticipated a small sample, which was appropriate for more in-depth qualitative questioning, but did not to conduct quantitative analyses using this group. To avoid confusion, we have removed reference to insufficient sample sizes.

Comment: Table 1a. Authors employed race and ethnicity, please, can explain these concepts in your research?

Response: We used the following race categories: non-Hispanic White, non-Hispanic Black, and non-Hispanic Other (including Asian or Pacific Islander, American Indian or Alaska Native, Other, and More Than One Race). We used the following ethnicity categories: Hispanic/Latino and non-Hispanic/Latino.

Comment: This type of device is more expensive than conventional cigarettes. Did you consider the socio-economic level of the study population?

Response: Thank you for this question. Although we did not ask respondents for their family’s household income, we did ask about lunch assistance program participation and government/public assistance. We dichotomized the sample into two income categories: low income vs. high income, with low income defined as participation in free/reduced-cost school lunch program or family receiving public assistance from government, both proxies for SES. Based on your question, we have added this variable into the model and made related changes in the manuscript.

Reviewer 2 Report

The present study uses a mixed-methods approach to evaluate the role of friends and socializing on vaping behaviors among adolescents. Due to increasing rates of vaping among this population, it is important to understand e-cigarette initiation and use behaviors. The authors have collected several types of data from a representative sample, and results have important implications for vaping policy and prevention efforts. I have several concerns about the manuscript that should be addressed prior to publication:

  1. To begin, I have concerns about the novelty of the study. The introduction provides an extensive overview of what is known about peer influence and vaping among adolescents. What does the present manuscript add that is not already known?
  2. I am curious to know the vaping characteristics of the sample for both studies. How many were current vapers? How many were vaping daily, or what did vaping rates look like? It might be interesting to test differences among those who experimented with e-cigarettes and those who continued to use.
  3. Elaborate on the specific methodology/software used for the “content analysis” of qualitative data.
  4. There is unnecessary text at the start of the results section that should be removed.
  5. Include more details about the weighing procedures used.
  6. The results section would benefit from being organized by study (Quant and Qual). As written, it jumps around between the two samples and it is difficult to interpret results within the context of the specific sample used.
  7. I suggest ranking the sources in Table 2 from most popular to least popular.
  8. Table 3 and Table 5 would be improved by adding a column/row that gave chi square and p values for significance testing.
  9. Provide justification for why self-esteem was measured and included in analysis. How does this relate to peer influence on vaping?
  10. To aid interpretation, significant results in Table 4 and Table 6 should be bolded.
  11. Figures 1 and 2 should have captions to describe the data presented. For instance, table 1 includes numbers in parentheses that are not described. Table 2 has circles of varying sizes, some overlapping, on different points of a line. Were these data driven images? What does overlap mean? Etc.
  12. The journal data is very interesting; however, these data are limited by the restricted presentation in the manuscript. These perhaps might be better presented in their own manuscript. For example, I am curious to know more about the vaping use patterns of the sample, the response rates (how many entries per day, how detailed, missing data and trends – did some start in the morning but lose interest later in the day?), and other themes that emerged. Although the manuscript emphasizes peer influence of vaping, many of the diary entries discuss situations where the adolescent is vaping alone or does not mention friends. This may have implications about those who transition from experimenting to regular use.
  13. The discussion section would benefit from suggestions from the authors about how to apply their findings in the rapidly changing environment of adolescent e-cigarette use. How can we use knowledge about peer relationships to leverage changes in social norms or other factors that can reduce e-cigarette initiation? What do we know about peer influence on cigarette smoking that we can draw from?
  14. As with other studies of e-cigarette use during a rapidly evolving product environment, limitations should address or propose future directions in regards to products that came on the market following data collection (eg. juul, puffbar).

Author Response

Thank you for the comments on our work. We appreciate your time and insights, and we believe that the draft has improved through the revision process. Below is a point-by-point response to the review comments, which addresses the ways that we revised the manuscript. Again, thank you for assisting us with improving the article.

Comment: The present study uses a mixed-methods approach to evaluate the role of friends and socializing on vaping behaviors among adolescents. Due to increasing rates of vaping among this population, it is important to understand e-cigarette initiation and use behaviors. The authors have collected several types of data from a representative sample, and results have important implications for vaping policy and prevention efforts. I have several concerns about the manuscript that should be addressed prior to publication:

Response: Thank you for your positive comments, especially on important implications for vaping policy and prevention efforts from our findings.

Comment: 

  1. To begin, I have concerns about the novelty of the study. The introduction provides an extensive overview of what is known about peer influence and vaping among adolescents. What does the present manuscript add that is not already known?

Response: We agree with your observation that the literature offers insight into factors that influence teen vaping. However, we approached the influence of friends from several angles, including friends as the source of initial ENDS use and added context in terms of where the initial use occurred. The findings clarify that, not only are teens influenced to vape by seeing their peers’ behavior, but their friends are supplying them with their first product, knowledge which is important in enforcing regulatory policies as well as in considering new policies.

Comment: 

  1. I am curious to know the vaping characteristics of the sample for both studies. How many were current vapers? How many were vaping daily, or what did vaping rates look like? It might be interesting to test differences among those who experimented with e-cigarettes and those who continued to use.

Response: Thank you for the suggestion to examine vaping status as a variable related to influence. We categorized ENDS users as current (35%), experimenters (21%), and former users (44%), based on definitions commonly employed in this literature. We found that experimenters have 1.6 times greater odds of saying their first vaping experience was hanging out with friends. Based on your comments, we added vaping status to the model. These results have been incorporated into the revised manuscript.

Comment: 

  1. Elaborate on the specific methodology/software used for the “content analysis” of qualitative data.

Response: Thank you for your comment. An inductive qualitative content analysis approach was used to identify themes that emerged from the data. Using an open coding technique, codes were assigned to responses using participants’ words and uploaded images to establish the coding scheme. We have added this information in the analysis section for our qualitative data.

Comment: 

  1. There is unnecessary text at the start of the results section that should be removed.

Response: Thank you for catching this error. Lines 165-167 were removed.

Comment: 

  1. Include more details about the weighing procedures used.

Response: Thank you for this suggestion. The data were weighted to be representative of the overall US population in terms of age, sex, race, ethnicity, and region based on U.S. Census data.

Comment: 

  1. The results section would benefit from being organized by study (Quant and Qual). As written, it jumps around between the two samples and it is difficult to interpret results within the context of the specific sample used.

Response: We appreciate this recommendation. We reorganized the results with the quantitative findings followed by the qualitative findings.

Comment: 

  1. I suggest ranking the sources in Table 2 from most popular to least popular.

Response: We reordered TABLE 2: Awareness of ENDS advertising or marketing as you suggested.

Comment: 

  1. Table 3 and Table 5 would be improved by adding a column/row that gave chi square and p values for significance testing.

Response: We added these values to tables 3 and 5.

Comment: 

  1. Provide justification for why self-esteem was measured and included in analysis. How does this relate to peer influence on vaping?

Response: We included self-esteem based on Social Learning Theory, which suggests that teens observe behaviors exhibited by their peers and adopt the behaviors that they perceive their peer group accepts. Having friends is associated with psychological well-being, which led to our curiosity about the association between peer influence and self-esteem. We found that the odds of ENDS users identifying a friend as the source of their first ENDS product were significantly higher among users with high self-esteem compared with users with low self-esteem.

Petraitis, J., Flay, B. R., & Miller, T. Q. (1995). Reviewing theories of adolescent substance use: Organizing pieces in the puzzle. Psychological Bulletin, 117(1), 67–86. https://doi.org/10.1037/0033-2909.117.1.67

Negriff S. Depressive symptoms predict characteristics of online social networks. J Adolesc Health. 2019 Jul;65(1):101-106. doi:10.1016/j.jadohealth.2019.01.026.

Comment: 

  1. To aid interpretation, significant results in tables 4 and Table 6 should be bolded.

Response: Thank you for the suggestion. We have bolded significant results in Tables 4 and 6.

Comment: 

  1. Figures 1 and 2 should have captions to describe the data presented. For instance, table 1 includes numbers in parentheses that are not described. Table 2 has circles of varying sizes, some overlapping, on different points of a line. Were these data driven images? What does overlap mean? Etc.

Response: We agree that the figures are enhanced with explanatory captions. We have added captions to Figure 1 (Numbers in parentheses represent the number of participants’ comments related to the theme) and Figure 2 (The size of the bubbles corresponds with the proportion of participants’ comments related to the theme).

We also revised captions and rearranged data for Table 2 to increase clarity.

Comment: 

  1. The journal data is very interesting; however, these data are limited by the restricted presentation in the manuscript. These perhaps might be better presented in their own manuscript. For example, I am curious to know more about the vaping use patterns of the sample, the response rates (how many entries per day, how detailed, missing data and trends – did some start in the morning but lose interest later in the day?), and other themes that emerged. Although the manuscript emphasizes peer influence of vaping, many of the diary entries discuss situations where the adolescent is vaping alone or does not mention friends. This may have implications about those who transition from experimenting to regular use.

Response: Thank you for your observation about the journal data. Based on your suggestion, we will address these data in more detail in a separate manuscript; thus, we have removed this information from the results.

Comment: 

  1. The discussion section would benefit from suggestions from the authors about how to apply their findings in the rapidly changing environment of adolescent e-cigarette use. How can we use knowledge about peer relationships to leverage changes in social norms or other factors that can reduce e-cigarette initiation? What do we know about peer influence on cigarette smoking that we can draw from?

Response: Thank you for your suggestion. We have added this language to the discussion section: Acknowledging the prominent role that social media play in many teens’ social networks, anti-vaping ads placed in social media platforms used by teens may have the potential to offset peer influence.

Comment: 

  1. As with other studies of e-cigarette use during a rapidly evolving product environment, limitations should address or propose future directions in regards to products that came on the market following data collection (eg. juul, puffbar).

Response: Thank you for pointing out this limitation. We have added it to the discussion of limitations.

Reviewer 3 Report

The Influence of Friends on Teen Vaping: A Mixed-methods 2 Approach

The objective of this study was to expand understanding of influences on teen vaping. The hypothesis is that friends play a role in teens vaping initiation. In general the concept of the manuscript is interesting, but the main problem is with the presentation of the data and comparison of two groups in terms of representation.

In the Abstract in Results Section not the percentages should be presented but the information derived from statistical models (crude and adjusted).

How these 39 ENDS user can be the representation of the 1549 ENDS user in the study? Are they similar in terms of demographic characteristics to the whole sample? Why those 39 took part in the follow up? What was their motivation to do it while the others no? If they are different than the w

Table 1a, b and Table 2 are just simple percentages. It will be more interesting to divided the tables in two groups males and females. What about different participant characteristics place of residence (urban, rural), lifestyle factors, smoking at home?

Table 3 and 5 is difficult to follow. It will be more clear to add the information about p-value in each line of the table. Maybe these two tables can be connected in one table.

Although the information received from the survey (39 ENDS users) is interesting how it can be translated in terms pf prevention. Some answers are only for 2-3 participants. Is it a real value? Additionally, citations of the answers in the figures is not the value in the scientific publications. 

Author Response

Thank you for the comments on our work. We appreciate your time and insights, and we believe that the draft has improved through the revision process. Below is a point-by-point response to the review comments, which addresses the ways that we revised the manuscript. Again, thank you for assisting us with improving the article.

Comment: The objective of this study was to expand understanding of influences on teen vaping. The hypothesis is that friends play a role in teens vaping initiation. In general the concept of the manuscript is interesting, but the main problem is with the presentation of the data and comparison of two groups in terms of representation.

In the Abstract in Results Section not the percentages should be presented but the information derived from statistical models (crude and adjusted).

Response: Thank you for this suggestion. We added the OR regarding social media in this sentence: In fact, the odds of the first vaping experience being with friends were 2 times higher for those who had seen ENDS ads on social media compared with other types of media.

Comment: How these 39 ENDS user can be the representation of the 1549 ENDS user in the study? Are they similar in terms of demographic characteristics to the whole sample? Why those 39 took part in the follow up? What was their motivation to do it while the others no? If they are different than the w

Response: The follow-up online community was fairly similar to the online survey sample in terms of gender (40-50% of each) and race and ethnicity (40-50% diverse populations vs. 50-60% non-Hispanic White). However, we have limited data about other characteristics of the online community participants. We have included this as a limitation.

Comment: Table 1a, b and Table 2 are just simple percentages. It will be more interesting to divided the tables in two groups males and females. What about different participant characteristics place of residence (urban, rural), lifestyle factors, smoking at home?

Response: Thank you for your suggestion. We expanded Table 1 to compare males and females across additional demographic variables. We also described data for place of residence and vaping status. Data on other lifestyle factors or passive smoking are not available in our survey.

Comment: Table 3 and 5 is difficult to follow. It will be more clear to add the information about p-value in each line of the table. Maybe these two tables can be connected in one table.

Response: Based on your suggestion, we have added p values to tables 3 and 5.

Comment: Although the information received from the survey (39 ENDS users) is interesting how it can be translated in terms pf prevention. Some answers are only for 2-3 participants. Is it a real value? Additionally, citations of the answers in the figures is not the value in the scientific publications.

Response: Thank you for your observation about the online community sample size. We have added this as a limitation.

Reviewer 4 Report

It is an interesting survey-study.

In the methods provide he inclusion criteria used for the survey and the following endpoints.

In table 1 data regarding second-hand smoke exposure and family history 

of smoking should be included . Were some comorbidities present , such as asthma?because asthma represents a controindication for smoking.

Please provide clarifications about table 1.b:Which sample is the table referring to?

Please provide informations on whether  people interviewed turned to a smoke-free center.

Please provide additional informations  on why breed comparisons were chosen within the subgroups.

In table 4 a column on statistical significance should be added.

Some adverse effects of e-cigarette vaping should be discussed such as EVALI syndrome and brocnhial inflammation.

I suggest to include some additional references about the effects of cigarette smoke and e-cigarettes on lung  and the pharmacologic and non pharmacologic approach and effect deriving from smoking cessation.

-J Emerg Med. 2020 Dec 11:S0736-4679(20)31354-8. doi: 10.1016/j.jemermed.2020.12.005

-J Comp Eff Res. 2013 May;2(3):335-43. doi: 10.2217/cer.13.25

-Am J Physiol Lung Cell Mol Physiol. 2018 Nov 1;315(5):L662-L672. doi: 10.1152/ajplung.00389.2017

-Subst Abus. 2018;39(3):289-306. doi: 10.1080/08897077.2018.1439802.

Author Response

Thank you for the comments on our work. We appreciate your time and insights, and we believe that the draft has improved through the revision process. Below is a point-by-point response to the review comments, which addresses the ways that we revised the manuscript. Again, thank you for assisting us with improving the article.

Comment: It is an interesting survey-study.

In the methods provide he inclusion criteria used for the survey and the following endpoints.

Response: Thank you for your positive comment. Per your suggestion, we have the inclusion criteria to the sample description.

Comment: In table 1 data regarding second-hand smoke exposure and family history of smoking should be included. Were some comorbidities present , such as asthma?because asthma represents a controindication for smoking.

Response: Our questionnaire did not include items on second-hand smoke exposure, family smoking history, or respondent health conditions. These would be valuable variables to include in future research.

Comment: Please provide clarifications about table 1.b:Which sample is the table referring to?

Response: We have renamed the table to specify the sample for our online community participants (TABLE 7: Qualitative online community participant characteristics-Analysis sample). Based on other revisions, the table also has been renumbered.

Comment: Please provide informations on whether people interviewed turned to a smoke-free center.

Response: The questionnaire did not include items regarding cessation. However, your point is a good one, and we hope to address this area in future research.

Comment: Please provide additional informations on why breed comparisons were chosen within the subgroups.

Response: Our analysis focused on comparisons by sex, age, race/ethnicity, and sexual orientation due to our research interest on populations vulnerable to tobacco use.

Comment: In table 4 a column on statistical significance should be added.

Response: Thank you for the suggestion. We have bolded significant results and included p values in Table 4.

Comment: Some adverse effects of e-cigarette vaping should be discussed such as EVALI syndrome and brocnhial inflammation. I suggest to include some additional references about the effects of cigarette smoke and e-cigarettes on lung and the pharmacologic and non pharmacologic approach and effect deriving from smoking cessation.

-J Emerg Med. 2020 Dec 11:S0736-4679(20)31354-8. doi: 10.1016/j.jemermed.2020.12.005

-J Comp Eff Res. 2013 May;2(3):335-43. doi: 10.2217/cer.13.25

-Am J Physiol Lung Cell Mol Physiol. 2018 Nov 1;315(5):L662-L672. doi: 10.1152/ajplung.00389.2017

-Subst Abus. 2018;39(3):289-306. doi: 10.1080/08897077.2018.1439802.

Response: Thanks for your suggestion. These are important aspects of vaping that our research team has addressed in separate publications; the scope of this analysis was limited to influencers of vaping initiation and not health consequences, but we did consider it important to acknowledge. We referenced the adverse health effects and added several of your suggested citations.